# A Rapid Water Region Reconstruction Scheme in 3D Watershed Scene Generated by UAV Oblique Photography

Yinguo Qiu [1],*, Yaqin Jiao [1], Juhua Luo [1], Zhenyu Tan [2], Linsheng Huang [3], Jinling Zhao [3], Qitao Xiao [1] and Hongtao Duan [1]

1   Key Laboratory of Watershed Geographic Sciences, Nanjing Institute of Geography and Limnology, Chinese Academy of Sciences, Nanjing 210008, China
2   College of Urban and Environmental Sciences, Northwest University, Xi'an 710127, China
3   National Engineering Research Center for Agro-Ecological Big Data Analysis & Application, Anhui University, Hefei 230039, China
*   Correspondence: ygqiu@niglas.ac.cn

**Abstract:** Oblique photography technology based on UAV (unmanned aerial vehicle) provides an effective means for the rapid, real-scene 3D reconstruction of geographical objects on a watershed scale. However, existing research cannot achieve the automatic and high-precision reconstruction of water regions due to the sensitivity of water surface patterns to wind and waves, reflections of objects on the shore, etc. To solve this problem, a novel rapid reconstruction scheme for water regions in 3D models of oblique photography is proposed in this paper. It extracts the boundaries of water regions firstly using a designed eight-neighborhood traversal algorithm, and then reconstructs the triangulated irregular network (TIN) of water regions. Afterwards, the corresponding texture images of water regions are intelligently selected and processed using a designed method based on coordinate matching, image stitching and clipping. Finally, the processed texture images are mapped to the obtained TIN, and the real information about water regions can be reconstructed, visualized and integrated into the original real-scene 3D environment. Experimental results have shown that the proposed scheme can rapidly and accurately reconstruct water regions in 3D models of oblique photography. The outcome of this work can refine the current technical system of 3D modeling by UAV oblique photography and expand its application in the construction of twin watershed, twin city, etc.

**Keywords:** oblique photography; 3D reconstruction; water region; twin watershed; real-scene 3D environment

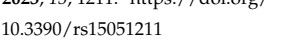



## 1. Introduction

Digital watershed technology has been considered the most powerful means for modern watershed planning and management. It can collect, represent and manage all kinds of watershed information by adopting synthetically several modern technologies, such as geographic information system (GIS), remote sensing (RS), virtual reality (VR), high-performance computing (HPC) [1–4], etc. Plenty of research has indicated that people can obtain more knowledge in 3D simulation scenes than in traditional 2D scenes [5–9]. For example, if one is in a virtual simulation scene, the impact of extreme weather can be understood more intuitively than if one were reading newspapers or watching TV programs. Consequently, the construction of 3D virtual simulative scenes of watersheds has received considerable attention from relevant scholars in the past two decades [10–12].

Early research on 3D visualization of watershed objects focused mainly on the 3D representation of watershed terrain, using digital elevation models (DEMs) and high-resolution remote sensing images [10,13,14]. Although 3D terrains of large areas can be represented rapidly by this kind of methods, the data volume of the generated 3D models is usually large, putting considerable pressure on data representation. The multi-resolution tile pyramid

technology was usually adopted to improve the rendering efficiency of 3D terrain data without affecting the visual effect [15,16]. Limited by scale, it is difficult for 3D terrain models generated based on DEMs to express the local detailed features of watersheds. As a result, detailed models of some key objects were commonly constructed manually and overlaid on the basis of a 3D terrain model [17,18]. To strike a balance between the fidelity and loading speed of 3D models, multiple models with various levels of detail were generally built for each entity and invoked on demand [19,20]. This problem can also be solved to a certain extent by classifying spatial entities according to their significance; the greater the importance of an entity, the higher the accuracy of its corresponding 3D models [21,22]. Desired effects of 3D visualization can be obtained with limited hardware conditions using the methods introduced in [19–22]. Nevertheless, high-precision models were mainly constructed manually, and the modeling processes were commonly inefficient. Moreover, traditional approaches to 3D visualization were mainly for man-made structures and objects, e.g., buildings, roads, etc., which are not suitable for modeling most categories of watershed factors, e.g., vegetation, farmlands and rivers.

In the past decade, 3D modeling technology based on UAV (unmanned aerial vehicle) oblique photography has developed rapidly, providing a new solution for rapid real-scene 3D reconstruction of watershed objects [23–25]. It collects the images of the target region from vertical and oblique angles simultaneously by configuring multiple sensors (cameras with five lenses are commonly used at present) on the same flight platform (UAV) [26]. Aerial triangulation is then implemented on the collected multi-view images to generate the total-factor 3D surface models of the target region, matching conjugate points in various multi-view images. Compared with traditional 3D modeling methods, oblique photography technology based on UAV has many advantages, e.g., high efficiency, low cost, strong authenticity [27–29], etc.

For the 3D modeling technology based on UAV oblique photography, the matching of the conjugate points in various multi-view images is a key step in reconstructing surface objects. Compared with other surface objects, such as buildings and water conservancy facilities, water regions have several unique features. Firstly, wind and waves are common phenomena in water regions, which usually result in various surface morphologies of the water region at different moments. Additionally, due to water reflection, the visual effects of water regions commonly differ from various angles of photography. As such, it is quite difficult to match conjugate points in various multi-view images when reconstructing water regions, and the obtained 3D models are usually irregular with many holes (Figure 1).

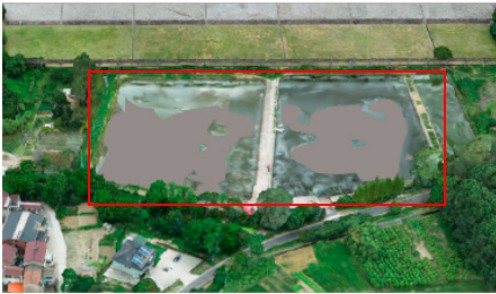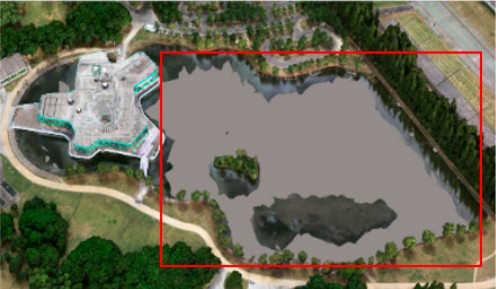

**Figure 1.** The effects of 3D reconstruction of water regions by UAV-based oblique photography technology, and the content in the red box corresponds to water regions in the real world.

In normal application scenarios, real information about water regions is not important and is thus seldom concentrated on, and virtual digital water-surface models are usually adopted to represent information of water regions in real-scene 3D environments [30–32]. Although the overall visual effect of the virtual scene is guaranteed, the real information about water regions, e.g., water color, floating objects, surrounding environment, etc., cannot be visualized. In some specific applications, e.g., water pollution management, water environment monitoring, etc., real information about water regions is significant for

decision-making, and traditional 3D watershed scenes cannot meet this demand. Based on high-resolution remote sensing images, the boundaries of water regions can be extracted using machine learning methods [33–37]. These methods, however, are only applicable to remote sensing images and cannot help in reconstructing water regions in 3D models of oblique photography. In addition, these methods can only extract the boundaries of water bodies and cannot obtain detailed texture information. In recent years, several commercial software programs, e.g., DP-Modeler [38], Meshmixer [39] and SVSModeler [40], have been developed to help repair the preliminary reconstruction results of water regions in 3D models of oblique photography. Nevertheless, a significant amount of human involvement is required when using these software programs, and the entire process of water region reconstruction is time-consuming. More significantly, in these methods, the texture of water-surface model is usually assigned through sample grabbing or texture interpolation, and rich texture information of water regions is still difficult to be represented in the 3D watershed scenes.

In summary, while UAV oblique photography technology has made it possible to rapidly and efficiently reconstruct multiple objects in a watershed, it cannot automatically and accurately obtain real information about water regions. As a result, information such as water color and floating objects is rarely represented in common 3D watershed scenes. The practicality of real-scene 3D models of watershed is not strong at present. To make a breakthrough in this field, a rapid reconstruction scheme for water regions in 3D models of oblique photography is proposed in this paper, the novelty of which can be summarized as follows:

(1)　A novel eight-neighborhood traversal algorithm has been designed and implemented. This algorithm can accurately and rapidly extract the boundary points of water regions in 3D models of oblique photography.

(2)　A fully automatic algorithm for texture image selection, preprocessing and mapping has been developed. This algorithm can intelligently map the textures of water regions based on the multi-view images acquired by UAV.

(3)　An evaluation system has been constructed for the reconstruction results of water regions in 3D models of oblique photography. This system can allow for both qualitative and quantitative evaluations of the reconstruction effect.

This paper is structured as follows: Section 2 describes the difficulties of reconstructing water regions based on oblique photography technology. Section 3 introduces the proposed rapid 3D reconstruction scheme for water region in detail. Finally, performance study and conclusions are presented in Sections 4 and 5, respectively.

## 2. Materials and Methods

### 2.1. Data Acquisition

In this study, a DJI M200 UAV equipped with five lenses was used to acquire multi-view images of the study area, which is Lake Tianmuhu watershed, a typical small watershed in the low mountains and hills in China. The main flight parameters of the used UAV were set as follows: flight altitude of 90 m, flight speed of 5 m/s, longitudinal overlap of 80% and sidelap of 60%. The main parameters of the camera used were as follows: sensor type CMOS, equivalent focal length of 24 mm, image resolution of $5472 \times 3078$, effective pixels of 20 million, 3 bands (red, green and blue) and calibrated IMU status. The orientations of the five lenses equipped in the used UAV remained constant during the data acquisition process, i.e., the orientation of the middle lens was vertically downward and the orientations of the other four lenses were all 45 degrees tilted inward. During the data acquisition process, images captured by different cameras were stored in independent paths.

After the acquisition of multi-view images, a software named "ContextCapture" was used to perform the aerial triangulation and generate the final 3D models of the research areas (in .obj format).

### 2.2. Boundary Point Extraction of Water Region

The definition of the spatial scope is the basis for the reconstruction of the water region. Considering that both the elevation and the density of point clouds of water regions in the 3D model of oblique photography are far lower than those of onshore areas, point density and point elevation are used as two constraints to identify the boundary points of the water regions in this section. The diagram of the boundary point extraction of the water region is shown in Figure 2. Given the original 3D model constructed using oblique photography technology, the process of the boundary point extraction of the water region can be demonstrated using the following seven steps:

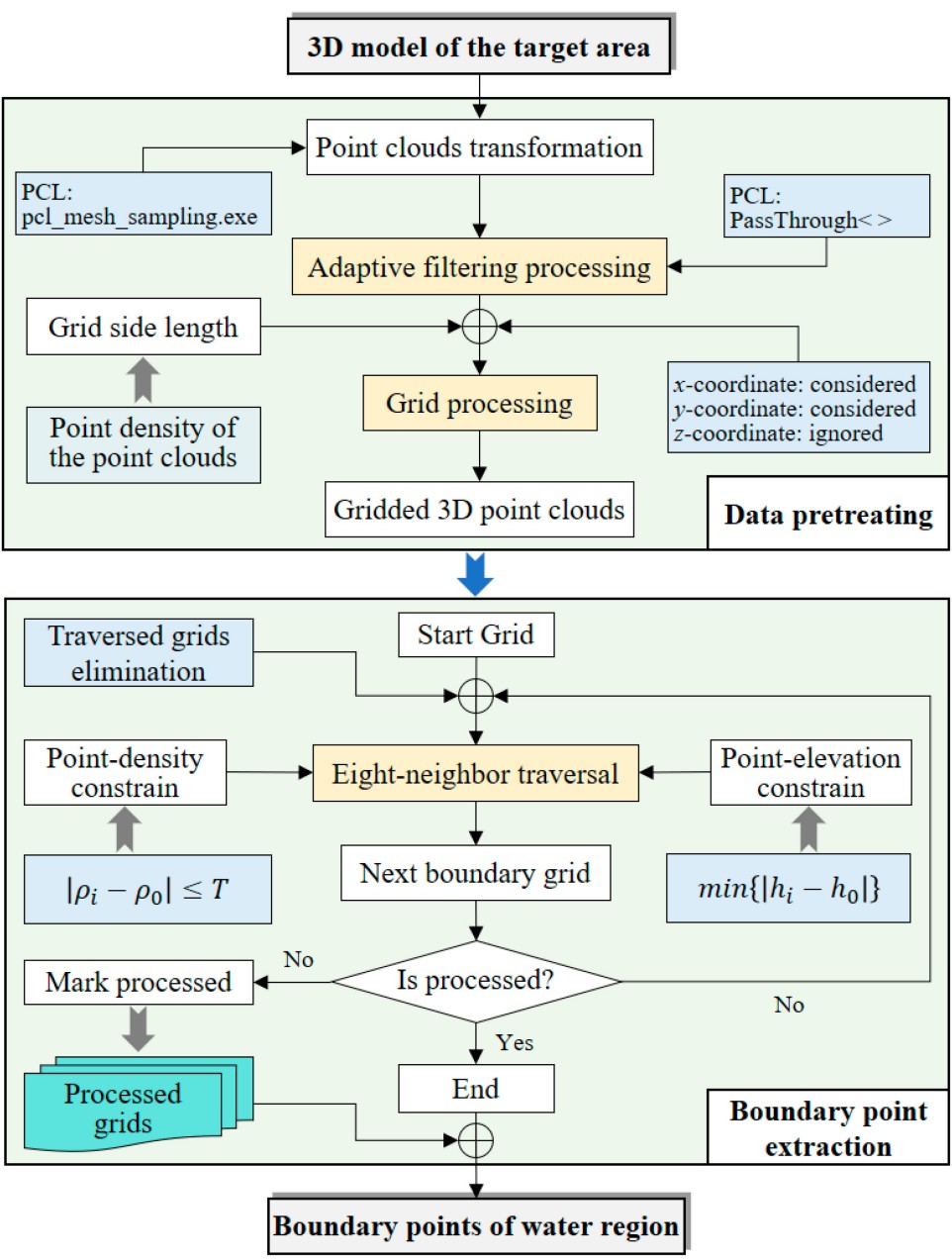

**Figure 2.** The flowchart of the procedure of boundary point extraction of water region.

Step 1. Transform the original 3D model into 3D point clouds. In this step, Point Cloud Library (PCL) [41], an open-source library for 2D/3D image and point cloud processing, is used to transform the 3D models obtained from oblique photography (in .obj format) into point clouds (in .pcd format). This step is designed to improve the generality of the

proposed method. However, if the original 3D point cloud data are available in specific applications, this step can be skipped.

Step 2. Eliminate interference points from the obtained 3D point clouds. Due to errors introduced during the 3D reconstruction process (specifically, the matching of conjugate points in various multi-view images as mentioned in the Section 1), there may be some erroneous points in the 3D point clouds with abnormal elevations (either too large or too little). To improve the efficiency of the proposed scheme, a path-through filter of point elevation based on PCL is constructed in this step to eliminate interference points, with the threshold values being determined adaptively based on the point density of the obtained 3D point clouds.

The analysis results of the 3D point clouds show that the point density at the top and bottom of the 3D point cloud data are significantly lower than in other areas. In this step, the threshold values of the path-through filter are determined based on the minimum (assumed e1) and maximum (assumed e2) elevations. The basis for this determination is that the densities of the points whose elevations are larger than e2 or less than e1 are all less than a given value.

Step 3. Divide the obtained 3D point clouds into independent grids. Construct a regular 2D grid and divide the obtained 3D point clouds into corresponding grids based on x and y coordinates of the 3D points. The side length of the constructed 2D grid is determined by the average point density of the obtained 3D point clouds. All the grids are initially marked as "unprocessed."

Step 4. Determine the starting grid for boundary extraction of the water region artificially. Note that the starting grid must be located on the boundary of the water region. The result of the starting point grid selection has an impact on the accuracy of water boundary extraction in theory. However, according to simulation experiments, the tolerance of the proposed scheme is satisfactory, and there is almost no influence on the result of boundary point extraction if the selected starting point grid is near the boundary of the water region (it does not need to be very precise).

Step 5. Search the adjacent boundary grid of the starting grid using the eight-neighbor analytical method (Figure 3). Point density and point elevation are two constraints used to identify the boundary point grid from the eight-neighbor grids. As shown in Figure 4, the grid marked as "S" is the starting grid, and its adjacent boundary grid will be searched from its eight adjacent grids (i.e., the grids marked as 0, 1, 2, 3, 4, 5, 6 and 7). The determined adjacent boundary grid must satisfy two conditions: (i) the difference in the point density of this grid and that of the starting grid is less than a given threshold, and (ii) the average value of point z-coordinates in this grid is the nearest one to that of the starting grid among the adjacent grids that satisfy condition (i).

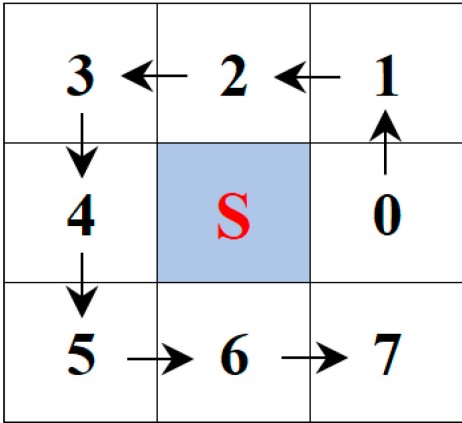

**Figure 3.** The flowchart of the eight-neighbor analytical method.

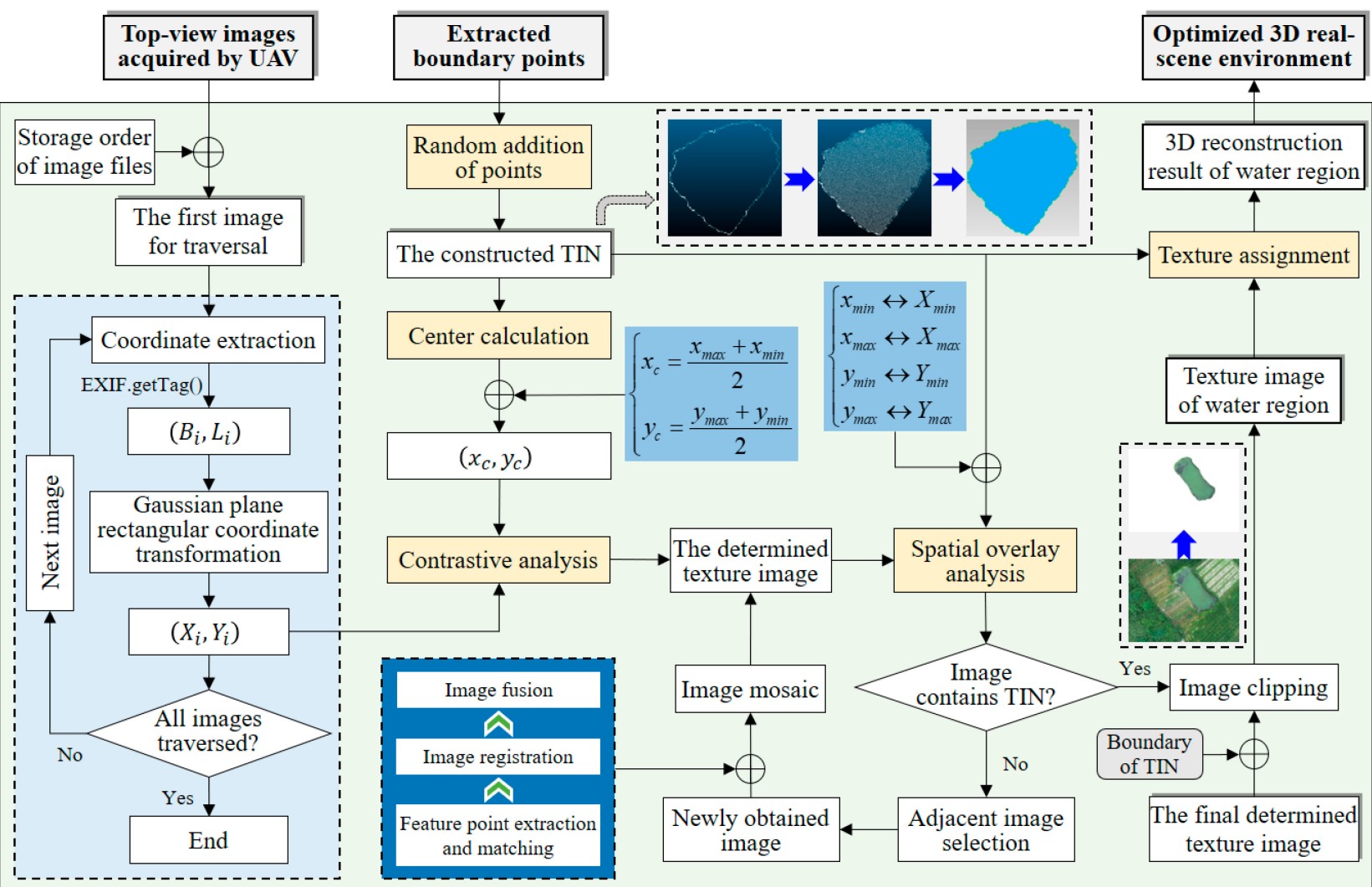

**Figure 4.** The flowchart of the procedure of TIN reconstruction and texture mapping of water region.

Step 6. If the identified boundary grid is marked as "processed", then terminate the process of boundary point extraction. Otherwise, mark the identified boundary grid as "processed," take it as the starting grid and repeat Step 5 (note that any grids that were traversed in the previous round of boundary grid searching will not be traversed again).

Step 7. Repeat Step 6 until the newly identified boundary grid is already marked "processed."

Finally, all points within the processed grids will be taken as the extracted boundary points of the water region. Using these extracted boundary points, the triangulated irregular network (TIN) model of the water region will be reconstructed, and then the corresponding texture image(s) will be processed and mapped to the constructed TIN, which will be introduced in detail in Section 2.3.

*2.3. Triangulated Irregular Network (TIN) Reconstruction and Texture Mapping of Water Region*

The diagram of TIN reconstruction and texture mapping of the water region is shown in Figure 4. After extracting the boundary points of water region, the TIN of the water region will be reconstructed, and the corresponding texture image will be automatically selected and mapped based on the following six steps:

Step 1. Reconstruct the TIN of the water region. Firstly, randomly add some points within the coordinate range of the extracted boundary points, with the number of the added points determined according to the coordinate range of the extracted boundary points as described in this paper. Then, construct the TIN based on the added points and the extracted boundary points using the Delaunay algorithm [42].

Step 2. Intelligently select the texture image of the constructed TIN. Both the reflection theory of light and practical experience indicate that the top-view images of the water region are least interfered with by reflections of objects on the shore. On this basis, a method of texture selection is designed and implemented in this paper, which can be summarized in the following sub-steps: (i) calculate the coordinates of the center of the extracted boundary points (denoted as $(x_c, y_c)$) using Equation (1), where $x_{max}$, $y_{max}$, $x_{min}$ and $y_{min}$ denote the maximum $x$-coordinate, the maximum $y$-coordinate, the minimum $x$-coordinate and the minimum $y$-coordinate of the extracted boundary points, respectively; and (ii) transform the geographical coordinates (represented by longitude and latitude) of each top-view image into rectangular coordinates (represented by x and y) by Equation (2), where $B$ and $L$ represent the latitude and the longitude of the image center, respectively; $N$ denotes the radius of curvature in prime vertical; $L_0$ represents the longitude of the Central Meridian; $a$ and $b$ are the major and minor axis semidiameters of the Earth's ellipsoid, respectively; $\rho$ is a constant with a value of 206,264.806247096355" and $X$ represents the ellipsoid arc length from the equator to the projection point of the image center on the reference ellipsoid [43]. Afterwards, all top-view images are traversed, the image whose coordinates are the nearest to $(x_c, y_c)$ is selected as the texture image.

$$\begin{cases} x_c = \frac{x_{max}+x_{min}}{2} \\ y_c = \frac{y_{max}+y_{min}}{2} \end{cases} \tag{1}$$

$$\begin{cases} x = X + \frac{N}{2\rho^2}\sin B \cos B l^2 + \frac{N}{24\rho^4}\sin B \cos^3 B\left(5 - \tan^2 B + \frac{9(a^2-b^2)}{b^2}\cos^2 B\right)l^4 \\ y = \frac{N}{\rho}\cos B l + \frac{N}{6\rho^3}\cos^3 B\left(1 - \tan^2 B + \frac{a^2-b^2}{b^2}\cos^2 B\right)l^3 + \frac{N}{120\rho^5}\cos^5 B(5 - 18\tan^2 B + \tan^4 B)l^5 \\ l = \frac{L-L_0}{\rho} \end{cases} \tag{2}$$

Step 3. Determine whether the TIN reconstructed in Step 1 is completely contained in the selected texture image. The inclusive relationship between the TIN and the texture image will be considered tenable only if the conditions listed in Equation (3) are all satisfied. In Equation (3), $x_{min}$ (resp. $x_{max}$) and $y_{min}$ (resp. $y_{max}$) represent the minimum (resp. maximum) $x$-coordinate and $y$-coordinate of points in the reconstructed TIN, while $X_{min}$ (resp. $X_{max}$) and $Y_{min}$ (resp. $Y_{max}$) represent the minimum (resp. maximum) $x$-coordinate and $y$-coordinate of the pixels in the selected texture image, respectively. The calculation

methods of $X_{min}$, $Y_{min}$, $X_{max}$ and $Y_{max}$ are shown in Equation (4), where $(X_i', Y_i')$ represents the center coordinate of the selected image, $W$ and $H$ are the width and height of the image, respectively, *Res* represents the pixel resolution of the image and $\lfloor \cdot \rfloor$ represents the rounding function.

$$\begin{cases} x_{min} \leq X_{min} \\ x_{max} \geq X_{max} \\ y_{min} \leq Y_{min} \\ y_{max} \geq Y_{max} \end{cases} \tag{3}$$

$$\begin{cases} X_{min} = X_i' - \left\lfloor \frac{W}{2} \right\rfloor \times Res \\ X_{max} = X_i' + \left\lfloor \frac{W}{2} \right\rfloor \times Res \\ Y_{min} = Y_i' - \left\lfloor \frac{H}{2} \right\rfloor \times Res \\ Y_{max} = Y_i' + \left\lfloor \frac{H}{2} \right\rfloor \times Res \end{cases} \tag{4}$$

Step 4. Determine the final texture image to be mapped to the TIN. If the inclusive relationship between the TIN and the texture image is tenable, the current selected image will be taken as the final texture image to be mapped. Otherwise, if the relationship is not tenable, the nearest image to the current image will be firstly selected from the top-view images based on coordinate comparison. Then, the two images will be merged to generate a mosaic image. The process of image mosaicing involves three main stages [44]: feature point extraction and matching, image registration, and image fusion. Afterwards, take the obtained mosaic image as the selected texture image and repeat Step 3 and Step 4 until the inclusive relationship between the TIN and the texture image is tenable.

Step 5. Map the final determined texture image to the reconstructed TIN of the water region. First, clip the selected texture image based on the vector boundary of the reconstructed TIN so that only the texture of the water region is retained. Then assign the clipped texture image to the reconstructed TIN, and the reconstructed 3D model of the water region can be obtained.

Step 6. Replace the original model data of the water region with the obtained 3D model. Delete all points within the boundary of the reconstructed TIN, and then place the reconstructed 3D model in the original model data.

After completing the steps described in Sections 2.2 and 2.3, the water region can be fully reconstructed, and the original 3D real-scene environment of the watershed can be optimized.

### 2.4. Accuracy and Effect Evaluation of Water Region Reconstruction

The 3D reconstruction of water regions in 3D models of oblique photography is currently a rarely researched topic, and thus effective indexes for evaluating the reconstruction results are lacking. In this paper, the corresponding index system is constructed to evaluate the reconstruction results of the water region, both qualitatively and quantitatively.

To evaluate the reconstruction results, qualitative comparisons will be made between the scene photos of the water regions and the screenshots of reconstructed results. Moreover, to further assess the reconstruction accuracy, boundary points of water regions in 3D models of oblique photography will be manually selected, and then an error analysis will be performed between the selected boundary points and the boundary points extracted by the proposed algorithm. Four indexes are adopted in this paper to assess quantitatively the reconstruction accuracy: average error (AE), root mean square error (RMSE), standard deviation (SD) and error of area (EOA).

### 3. Experiments and Results

Ten 3D models generated by UAV oblique photography technology in Figure 5 (in .obj format) are used in this section as experimental data to test the performance of the proposed 3D reconstruction scheme for water regions. Table 1 lists the basic properties of the ten experimental 3D models, i.e., number of points and coordinate ranges. In this section, the

experiments were conducted on a PC with a configuration: CPU Inter Core i7-8700 3.20 GHz, GPU Intel® UHD Graphics 630, RAM 64 GB and OS Windows 10 Education (×64).

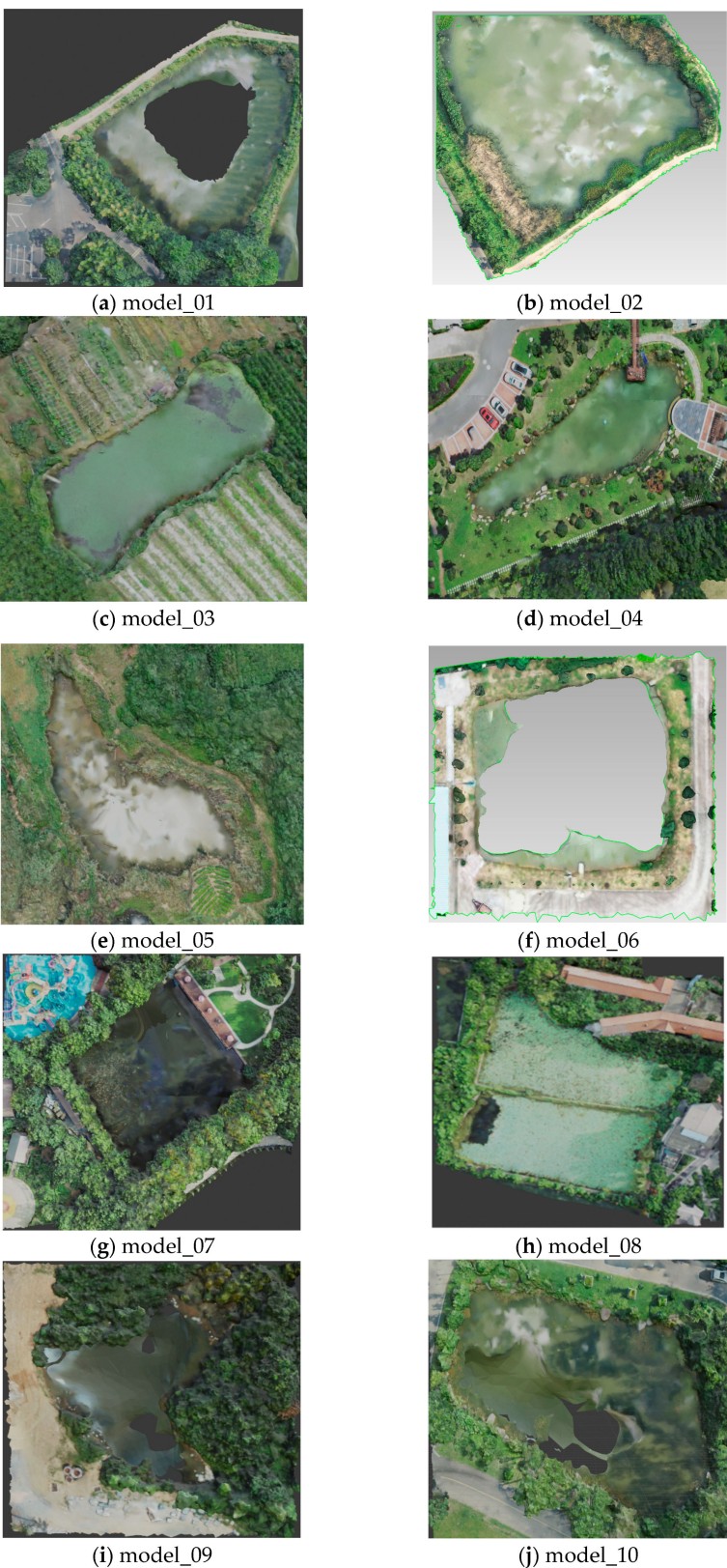

**Figure 5.** The experimental 3D models of oblique photography.

**Table 1.** Properties of the experimental 3D models.

| 3D Model | Number of Points | Coordinate Range ($x$) | Coordinate Range ($y$) | Coordinate Range ($z$) |
|---|---|---|---|---|
| model_01 | 185,581 | −83.8019~25.1597 | 196.6981~293.4608 | 15.4556~40.8227 |
| model_02 | 55,318 | −83.8019~0.0464 | 199.0290~330.9512 | 20.4495~36.4812 |
| model_03 | 76,291 | −98.4907~50.1999 | 516.4602~588.6694 | 65.4179~84.0148 |
| model_04 | 60,339 | 325.1720~369.3640 | −159.8860~−106.2403 | 38.4707~53.2799 |
| model_05 | 170,346 | 10.6901~107.8690 | −130.7712~−48.9058 | 95.0714~110.2020 |
| model_06 | 18,701 | −15.1823~26.1760 | 303.8401~338.9920 | 125.6604~134.7461 |
| model_07 | 372,732 | −158.1614~−32.8305 | −153.5332~−31.5963 | 38.0605~69.0925 |
| model_08 | 348,132 | −523.1904~−377.0732 | 52.6536~223.9019 | 20.9784~67.3059 |
| model_09 | 72,168 | 32.6015~93.5034 | −262.7206~−200.4312 | 77.8785~97.9674 |
| model_10 | 120,951 | −157.7250~−93.8586 | −37.0853~26.7636 | 19.6439~52.9920 |

As described in Step 1 of Section 2.2, the ten experimental 3D models of oblique photography were first transformed into 3D point clouds (in .pcl format). The results of the transformation are shown in Figure 6. Afterwards, interference points were eliminated from the obtained 3D point clouds, and all 3D point clouds were then divided into independent grids, as mentioned in Step 2 and Step 3 of Section 2.2. Then, the boundary points of water regions were extracted from the ten pretreated 3D point clouds, using the method explained in Step 4, Step 5, Step 6 and Step 7 of Section 2.2. The results of boundary point extraction of water regions are shown in Figure 7.

As mentioned in Section 2.3, after extracting the boundary points of water regions of the ten experimental 3D models, TINs of water regions were then constructed, and texture images were automatically selected and mapped. Afterwards, the original model data of water regions were replaced by the reconstructed 3D models. The final results of 3D reconstruction of water regions are shown in Figure 8.

### 3.1. Qualitative Evaluation

It can be seen from Figure 5 (the original 3D models of oblique photography) and Figure 8 (the final results of the water region reconstruction) that water regions in the original 3D models of oblique photography can be effectively reconstructed. Holes can be filled, and the real information of water regions can be represented in the real-scene 3D environments, thanks to the proposed scheme.

In order to further assess the reconstruction effect of water regions, visual comparisons were conducted between the final obtained 3D models of water regions and the scene photos captured by UAV. The comparison results indicate that the reconstruction effect of water regions is visually good. Taking one of the ten experimental 3D models as an example, the comparison results are shown in Figure 9.

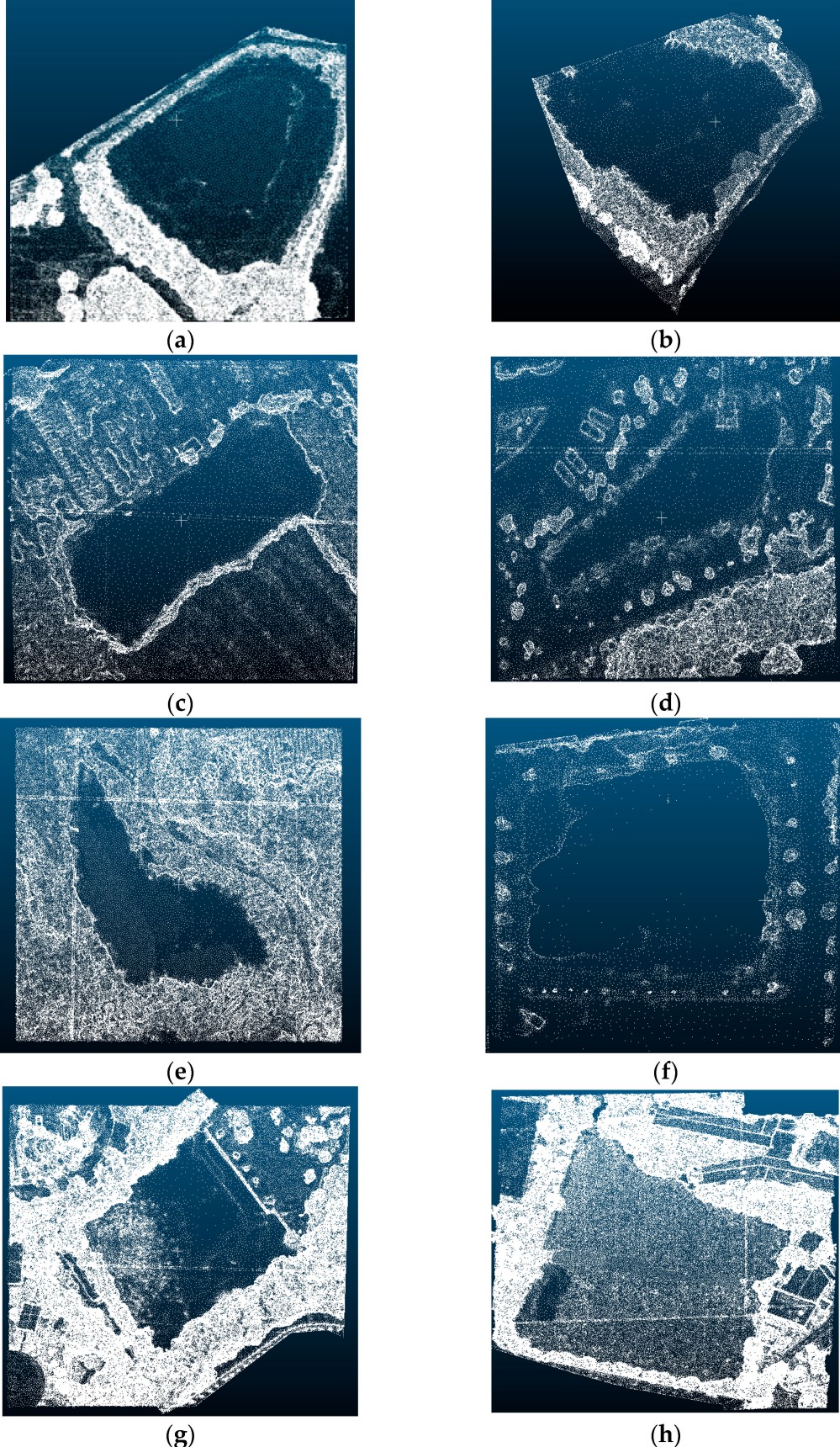

**Figure 6.** *Cont.*

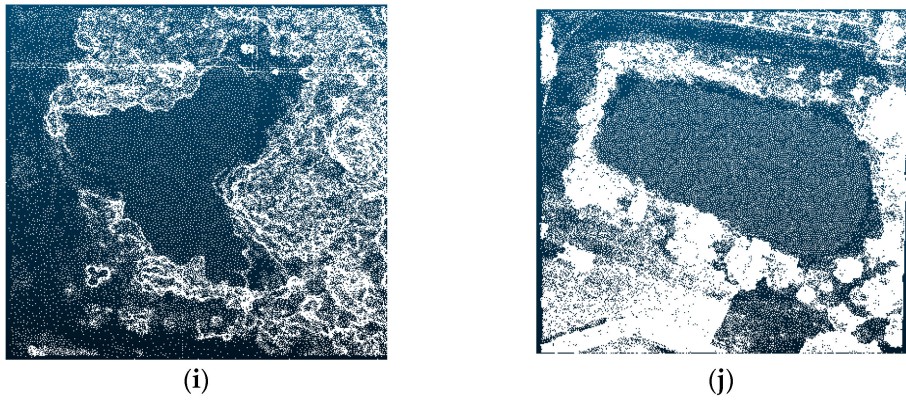

**Figure 6.** Results of 3D point clouds transformation of the ten experimental 3D models. (**a–j**) are the 3D point clouds of model_01, model_02, model_03, model_04, model_05, model_06, model_07, model_08, model_09 and model_10, respectively.

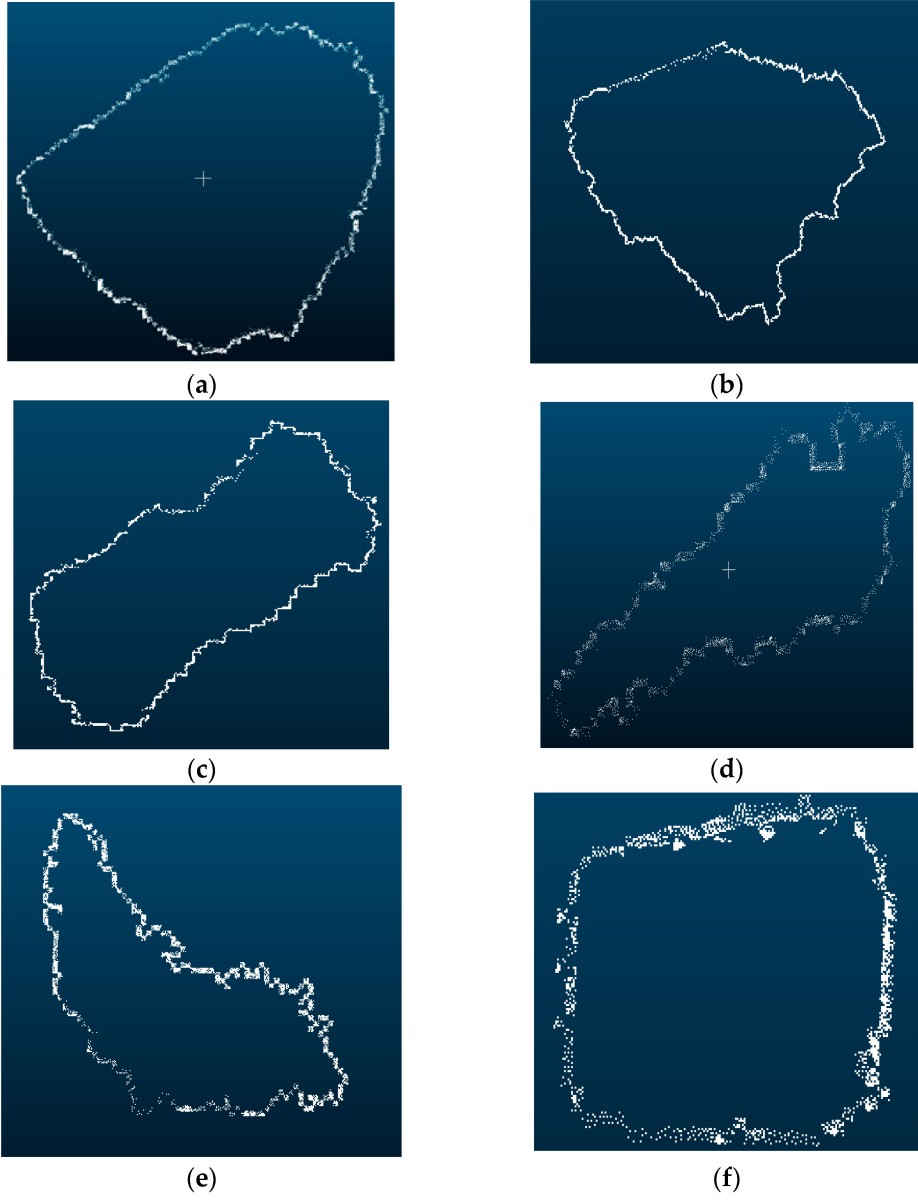

**Figure 7.** *Cont.*

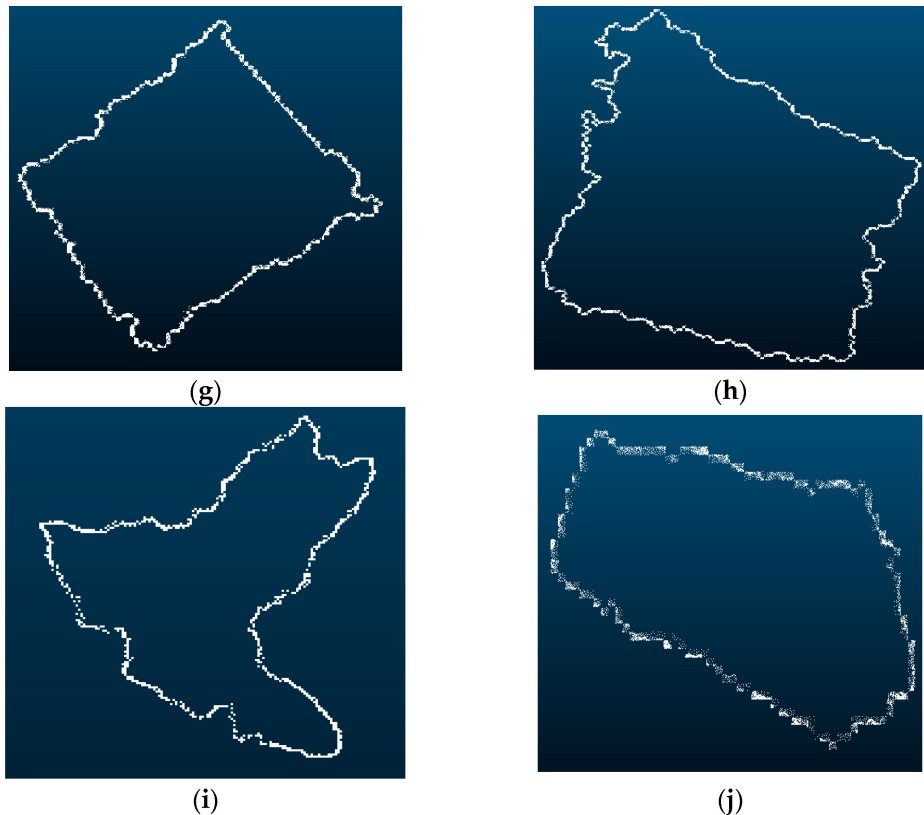

**Figure 7.** Results of boundary point extraction of the ten experimental 3D models. (**a–j**) are the results of boundary point extraction of water regions in model_01, model_02, model_03, model_04, model_05, model_06, model_07, model_08, model_09 and model_10, respectively.

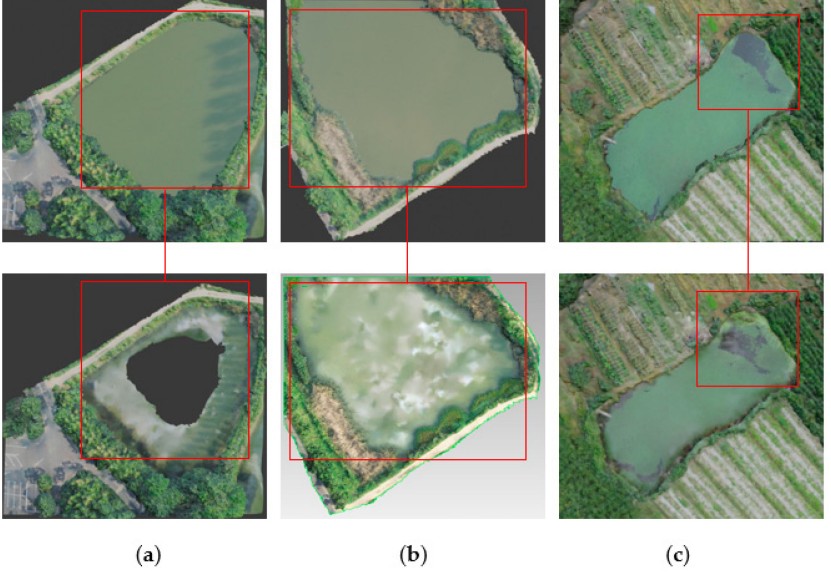

**Figure 8.** *Cont.*

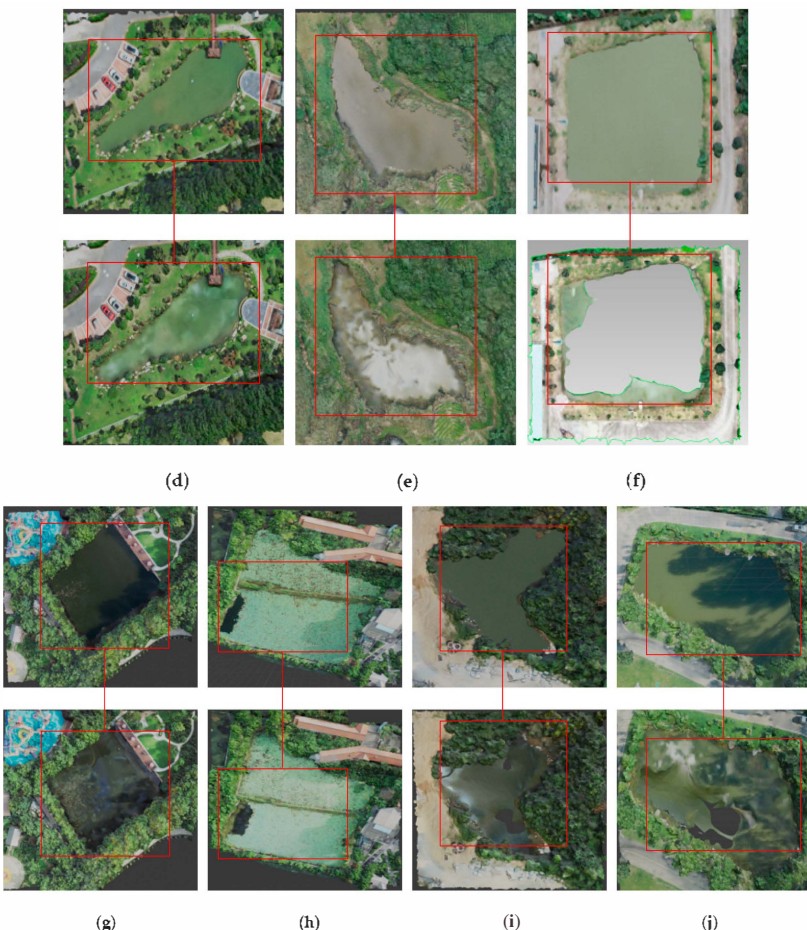

**Figure 8.** Results of water region reconstruction. (**a**–**j**) are the reconstruction results of water regions in model_01, model_02, model_03, model_04, model_05, model_06, model_07, model_08, model_09 and model_10, respectively. For each result, the upper image is the reconstruction result, the bottom image is the original 3D model of photography, and the framed regions are the effects of local alignment.

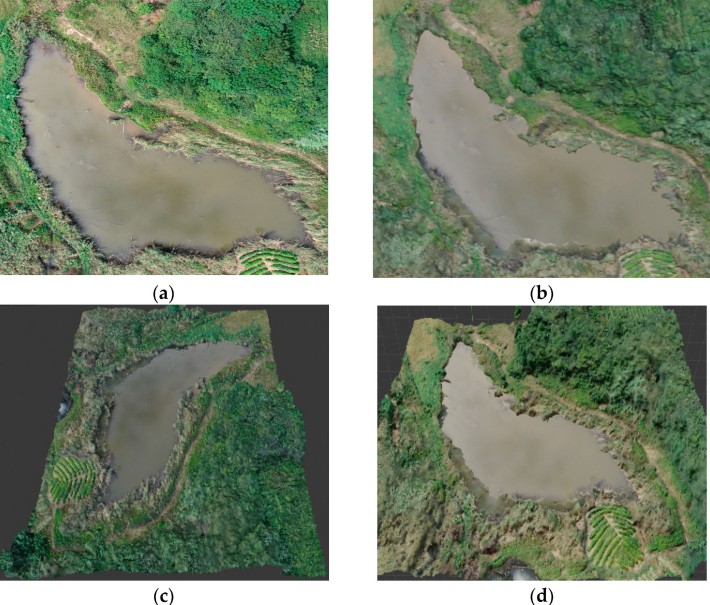

**Figure 9.** Comparison results of reconstruction effect of water regions. (**a**) is the scene photo captured by UAV, and (**b**–**d**) are the screenshots of the reconstruction result from three different angles.

*3.2. Quantitative Evaluation*

In addition to qualitative assessments, quantitative evaluations have also been carried out in this paper. Firstly, an accuracy comparison was made among this scheme and two state-of-the-art works, i.e., schemes in [33,37], in terms of water boundary extraction. The accuracy of each scheme was evaluated based on the manually selected boundary points of water regions using the four indexes explained in Section 2.4. Moreover, tests were performed with a general hardware configuration to evaluate the efficiency of the proposed scheme.

(1)    Accuracy Evaluation

For each experimental 3D model, the water boundary was firstly manually depicted, and then extracted using the proposed scheme. The corresponding texture image generated in Step 4 of Section 2.3 was also adopted for the water boundary extraction of the schemes in [33,37]. Afterwards, the point coordinate errors of this scheme and the schemes in [33,37] were calculated and analyzed based on the manually depicted water boundary, using the four indexes explained in Section 2.4, i.e., *AE*, *RMSE*, *SD* and *EOA*. The experimental results are shown in Table 2. It can be seen from Table 2 that the average values of AE, RMSE and SD of this scheme are all less than 0.6 m, and the average value of EOA is less than 4%. In addition, the accuracy of the water boundary extraction of the proposed scheme is acceptable compared with state-of-the-art works.

**Table 2.** Comparison results of accuracy of water boundary extraction.

| 3D Model | Schemes | AE (Unit: m) | RMSE (Unit: m) | SD (Unit: m) | EOA (Unit: %) |
|---|---|---|---|---|---|
| model_01 | This scheme | 0.293671 | 0.436409 | 0.246782 | 1.162059 |
| | Scheme in [33] | 0.502648 | 0.623545 | 0.532614 | 5.369245 |
| | Scheme in [37] | 0.456841 | 0.598413 | 0.508592 | 6.326541 |
| model_02 | This scheme | 0.154879 | 0.342610 | 0.107922 | 0.070159 |
| | Scheme in [33] | 0.559348 | 0.657246 | 0.501236 | 4.895633 |
| | Scheme in [37] | 0.585645 | 0.623598 | 0.486235 | 5.632154 |
| model_03 | This scheme | 0.470261 | 0.613727 | 0.297445 | 2.343532 |
| | Scheme in [33] | 0.523648 | 0.641235 | 0.526354 | 5.553241 |
| | Scheme in [37] | 0.513647 | 0.623194 | 0.543969 | 5.763215 |
| model_04 | This scheme | 0.302912 | 0.399930 | 0.595390 | 2.802301 |
| | Scheme in [33] | 0.469216 | 0.615623 | 0.586324 | 5.230684 |
| | Scheme in [37] | 0.521853 | 0.612548 | 0.572694 | 4.796523 |
| model_05 | This scheme | 0.891332 | 0.797488 | 0.676160 | 7.293576 |
| | Scheme in [33] | 0.493257 | 0.665123 | 0.563247 | 5.639521 |
| | Scheme in [37] | 0.462584 | 0.691254 | 0.586218 | 5.326566 |
| model_06 | This scheme | 0.392715 | 0.567713 | 0.236397 | 6.951579 |
| | Scheme in [33] | 0.562415 | 0.665238 | 0.563241 | 5.845524 |
| | Scheme in [37] | 0.512398 | 0.698423 | 0.543697 | 6.320227 |
| model_07 | This scheme | 0.272851 | 0.458527 | 0.185233 | 4.731892 |
| | Scheme in [33] | 0.521563 | 0.652347 | 0.536247 | 5.369656 |
| | Scheme in [37] | 0.486325 | 0.642359 | 0.512398 | 5.785544 |
| model_08 | This scheme | 0.389727 | 0.488049 | 0.344092 | 1.054049 |
| | Scheme in [33] | 0.554236 | 0.631251 | 0.563244 | 4.763565 |
| | Scheme in [37] | 0.542169 | 0.657358 | 0.523692 | 5.221526 |
| model_09 | This scheme | 0.677022 | 0.707902 | 0.489006 | 11.141575 |
| | Scheme in [33] | 0.516974 | 0.684592 | 0.553622 | 5.912548 |
| | Scheme in [37] | 0.536958 | 0.645963 | 0.543669 | 5.632411 |
| model_10 | This scheme | 0.221631 | 0.405329 | 0.160023 | 1.247305 |
| | Scheme in [33] | 0.542354 | 0.676523 | 0.523687 | 6.231521 |
| | Scheme in [37] | 0.523627 | 0.645286 | 0.526384 | 5.454217 |
| Average of ten models | This scheme | 0.406700 | 0.521768 | 0.333845 | 3.879803 |
| | Scheme in [33] | 0.5245659 | 0.6512723 | 0.5449816 | 5.4811138 |
| | Scheme in [37] | 0.5142047 | 0.6438396 | 0.5347548 | 5.6258924 |

(2)　Efficiency Evaluation

For each experimental 3D model, the consumed time of 3D reconstruction of the water region was recorded and is presented in Table 3 (the configuration of the used PC is described in the first paragraph of Section 3). The consumed time includes the entire process of water region reconstruction, except the manual determination of the start-grid for boundary extraction of water regions, i.e., Step 4 of Section 2.2. It can be concluded from Table 3 that the proposed scheme is quite efficient.

**Table 3.** Results of efficiency evaluation of this scheme.

| 3D Model | Consumed Time (Unit: s) |
|---|---|
| model_01 | 10.851 |
| model_02 | 5.053 |
| model_03 | 8.833 |
| model_04 | 6.262 |
| model_05 | 7.522 |
| model_06 | 18.135 |
| model_07 | 34.599 |
| model_08 | 33.561 |
| model_09 | 16.542 |
| model_10 | 31.755 |
| Average of ten models | 17.311 |

## 4. Discussion

A novel, rapid and accurate reconstruction scheme for water regions in 3D models of oblique photography is proposed in this paper. Accurately reconstructing the water region in 3D models of oblique photography poses two main problems: (i) accurately modeling the water surface, and (ii) real-texture mapping of the reconstructed water surface model.

For the first problem, it is assumed that the surfaces of water regions are flat. In this case, the modeling of water surface can be transformed into the boundary extraction of the water region. The distinct point cloud characteristics (i.e., elevation and density) of water regions and other objects provide the foundation for the accurate identification of the boundary of land and water regions. On this basis, an algorithm for the boundary point extraction of the water region in 3D models of oblique photography is designed in this paper (explained in Section 2.2). Experimental results indicate that the accuracy of boundary extraction results can be at the centimeter level.

For the second problem, both the reflection theory of light and practical experience indicate that the top-view images of water regions are least interfered with by reflections of objects on the shore. In this case, the real-texture mapping of the reconstructed water-surface model can be transformed into the intelligent optimization and processing of top-view images. In fact, the metadata of each image captured by digital cameras are recorded and bound to themselves (named EXIF information). The content of EXIF contains image coordinates, image resolution, shooting time, etc. On this basis, a method of texture selection is designed and implemented in this paper. Furthermore, there was a situation where the spatial range of the selected top-view image cannot completely contain the target water region. To solve this problem, another method was designed in this paper (mentioned in Step 3 of Section 2.3). Experimental results have verified the effectiveness of the proposed methods.

In summary, the scheme presented in this paper can improve the current technical system of 3D modeling by UAV oblique photography. The outcomes of this paper can contribute to the construction of real-scene 3D environments in many application fields, e.g., twin watershed, twin lakes, twin city, etc. Meanwhile, there are four aspects of work that are valuable to be further researched in future studies. Firstly, in this work, the starting grid for boundary point extraction is determined manually, which is the only step requiring manual intervention in this scheme. Therefore, automatic determination methods for the

starting grid should be studied further in the near future. Secondly, in the proposed scheme, both the TIN and the corresponding texture image are only processed and represented at the finest level, and the LOD (levels of detail) technology is not applied to improve the rendering efficiency of 3D scenes. In future work, more attention should be paid to the generation of TINs and texture images of water regions with various levels of fineness based on LOD technology to obtain an efficient rendering operation. Thirdly, the texture information is obtained only based on the top-view images of UAV in this paper, and there are some interfering factors in the final generated texture images of water regions, especially shadows of objects on the shore. Thus, there is still much room for improvement in texture mapping. In future studies, it is worthy to further develop intelligent algorithms for detecting and eliminating interfering factors in the texture images of water regions. Finally, it is assumed in this paper that the surfaces of water regions are flat, and thus objects within water regions, such as islands, towers, etc., cannot be 3D reconstructed, which is worthy of further study.

## 5. Conclusions

UAV oblique photography technology is providing a new and efficient solution for the 3D reconstruction of ground objects on a watershed scale. Although real-scene 3D environments can be established rapidly, water regions cannot be effectively reconstructed using this technology and real information about water regions is rarely visualized in 3D watershed scenes. As such, traditional 3D simulation environments of watersheds cannot meet the increasing demands of integrated watershed management.

To address the aforementioned problem, this paper proposes a rapid 3D reconstruction scheme for water regions in 3D models of oblique photography. Firstly, boundary points of the water region are extracted using a designed eight-neighbor traversal-based algorithm. Next, the TIN of the water region is constructed using the Delaunay algorithm. Afterwards, the corresponding texture image is intelligently selected and automatically processed. Finally, the processed texture image is mapped to the TIN to obtain the reconstructed 3D model of the water region. Simulation experiments have shown that the proposed scheme is accurate, efficient and effective.

In future studies, further research should be conducted on automatic determination methods for the starting grid, the generation of TINs and texture images of water regions with varying levels of fineness based on LOD technology, the detection and elimination of interfering factors in the texture images of water regions and the 3D reconstruction of objects within water regions.

**Author Contributions:** Conceptualization, Y.Q., H.D., J.L. and Y.J.; methodology, Y.Q., L.H. and J.Z.; software, Y.J.; validation, Y.J., Z.T. and Q.X.; formal analysis, Y.Q.; investigation, Y.J.; resources, H.D.; data curation, Y.J.; writing—original draft preparation, Y.Q.; writing—review and editing, H.D.; visualization, J.L.; supervision, Q.X.; project administration, Y.Q.; funding acquisition, Y.Q., H.D. and J.L. All authors have read and agreed to the published version of the manuscript.

**Funding:** This research was funded jointly by the Natural Science Foundation of Jiangsu Province (Grant No. BK20201100), the National Natural Science Foundation of China (Grant No. 42101433, 41971309 and 41971314) and the Open Research Fund of National Engineering Research Center for Agro-Ecological Big Data Analysis & Application, Anhui University (Grant No. AE202107).

**Data Availability Statement:** The data and materials that support the findings of this study are freely available upon request from the corresponding author at the following e-mail address: ygqiu@niglas.ac.cn.

**Acknowledgments:** We would like to acknowledge the assistance of Jia Liu and Xiaokang Ding in the design and evaluation of simulation experiments.

**Conflicts of Interest:** The authors declare no conflict of interest.

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
