# Peer review of "A Rapid Water Region Reconstruction Scheme in 3D Watershed Scene Generated by UAV Oblique Photography"

_remotesensing, doi:10.3390/rs15051211_

Round 1

Reviewer 1 Report

Line 13: Geographical objects is more relevant.

Line 74: The science of 3D reconstruction from images is clear, there is no need to include a Figure to show it. relevant references are enough.

Why not use water indices to detect, map and then include water regions in the final data? 

I am not sure authors are addressing water regions i.e., the area covered by water, or on the contrary, they are attempting to map water levels. The information on water levels of waterbodies seems more interesting and appealing from a management perspective.

Material and methods

The authors do not provide detail about camera models and their specifications in the article. This is important information to incorporate in an article for others to obtain and reconstruct similar results. Having said that, please include the following camera-related details. Sensor type, resolution, number of bands, calibration status, etc. what was the camera orientation in space and time e.g., Nadir, off-Nadir pointing etc?

Line 145: Most of the photogrammetry software packages provide point clouds as the primary product along with DSM and DEM. Why authors need to process and transform a 3D model into 3D point clouds.

Interference point elimination? what are they?

At this point, I have two observations that need to address in the revised version.

First, the related work sub-section to show similar studies done in past and their drawback and conclusions.

Second, using high-resolution images, water boundaries can be extracted using state-of-the-art machine learning methods about which authors do not provide any relevant information.

Third, the authors do not clearly mention either its water regions or water levels to be mapped.

Results.

The water surface status and statistics or the current status of water characteristics such as floating objects, watercolor, and its variations, and texture seem to be gone from the final results, which is indeed useful information for managing and monitoring the open water bodies.  Figure 9, therefore, shows false information after reconstruction. How authors address this as real information is replaced with a uniform water texture. 

Also, when the water surface has texture in images then reconstruction using photogrammetry methods should not be a problem.

If the authors can address all these findings in the revised version, I hope this can be published. 

hsduo and

Reviewer 2 Report

Great paper, water has been an issue for many in the photogrammetry space.

I would have included some of the work from NERF based modelling, but without any known dimensions in the model, it would only show that water can be modelled in object space, just not with real world coordinates at the moment.

Reviewer 3 Report

The paper develops an optimization scheme for water region 3D models of oblique photography. Overall, the topic is interesting and of practical value. The paper’s structure is logical and fluent. However, there are some issues that need to be addressed before the paper can be suggested for publication.

(1)   Introduction: the problem statement should be more concise, i.e., paragraph 2 seems unnecessary. The picture is expanded too large and suggests focus on the current progresses and difficulties of 3d reconstruction of water region based on UAV oblique photography, which appears in the 3rd paragraph. 

(2)   Methodology: if I understand correctly, the scheme extracts the outer boundary of water region, and then fills the region with a top-view image. Then what if there is indeed islands or other feature in the water region?

Step 2 (line 150). What is the relationship between threshold and point density?

Step 4 (line 158). Does it mean the start point selection affects the quality of output model? Is tolerance taken into consideration?

2.4 equation 5. these are well-known equations and suggest not presenting them.

(3)   Experiments and Results: Please compare the proposed scheme with other current methods in terms of accuracy and efficiency. Does the artificial point selection determine the accuracy? 

(4)   Discussion: suggest moving the first paragraph to the introduction. And the discussion session reads like a conclusion or abstract, please add future studies.

(5) There are some typos and grammar errors, please double-check throughout the paper. 

Round 2

Reviewer 1 Report

Dear authors,

Thank you for improving the quality and content of this article. Article has improved to be considered for publication. 

Congratulations.

Reviewer 3 Report

I appreciate the authors' efforts in addressing previous comments. The revision is decent and the paper quality has been improved. A further suggestions is as follows:

Experiments and results: please compare the proposed scheme with other current methods.

Although the author claim the work is an exploring rescharch and there is no relevant research at present.  Each portion of the scheme (from boundary extraction to texture image selection), a number of algorithms are available in the community. Without a comparison, it is not persuasive enough to say that the proposed method is better. 
